# Effects of Surface Plasmon Coupling on the Color Conversion of an InGaN/GaN Quantum-Well Structure into Colloidal Quantum Dots Inserted into a Nearby Porous Structure

**DOI:** 10.3390/nano13020328

**Published:** 2023-01-12

**Authors:** Shaobo Yang, His-Yu Feng, Yu-Sheng Lin, Wei-Cheng Chen, Yang Kuo, Chih-Chung (C. C.) Yang

**Affiliations:** 1Institute of Photonics and Optoelectronics, and Department of Electrical Engineering, National Taiwan University, No. 1, Section 4, Roosevelt Road, Taipei 10617, Taiwan; 2Department of Energy and Refrigerating Air-Conditioning Engineering, Tungnan University, 152 Beishen Road, Section 3, New Taipei City 22202, Taiwan

**Keywords:** surface plasmon coupling, color conversion, colloidal quantum dot, Förster resonance energy transfer, porous structure, nanoscale-cavity effect

## Abstract

To further enhance the color conversion from a quantum-well (QW) structure into a color-converting colloidal quantum dot (QD) through Förster resonance energy transfer (FRET), we designed and implemented a device structure with QDs inserted into a GaN nano-porous structure near the QWs to gain the advantageous nanoscale-cavity effect. Additionally, surface Ag nanoparticles were deposited for inducing surface plasmon (SP) coupling with the QW structure. Based on the measurements of time-resolved and continuous-wave photoluminescence spectroscopies, the FRET efficiency from QW into QD is enhanced through the SP coupling. In particular, performance in the polarization perpendicular to the essentially extended direction of the fabricated pores in the nano-porous structure is more strongly enhanced when compared with the other linear polarization. A numerical simulation study was undertaken, and showed consistent results with the experimental observations.

## 1. Introduction

Photon color conversion is a useful process for color display application. The performance of such a process relies on the efficiencies of two mechanisms, including the energy transfer from the energy donor into the acceptor and the acceptor emission. If the distance between the donor and acceptor is smaller than several tens nm, the Förster resonance energy transfer (FRET) can lead to a high efficiency of energy transfer [1]. Based on the electromagnetic theory, FRET can be regarded as a process of the acceptor absorption of the near-field energy produced by the donor [2,3]. In other words, donor energy is effectively transferred into the acceptor without the far-field process of photon emission-absorption. The near field distribution produced by the donor is affected by its surrounding structure. When the donor is placed near a nanoscale cavity, the near field distribution in a certain portion of the cavity can be enhanced [4]. Therefore, by placing the acceptor inside the nanoscale cavity, the energy absorbed by the acceptor or FRET efficiency can be enhanced [4,5]. In such a structure, the emission efficiency of the acceptor can also be increased through the nanoscale-cavity effect. The increased emission efficiency of a light emitter inside a nanoscale cavity is caused by the modification of the near field distribution through the Purcell effect [6], which can enhance the far-field radiated power. Hence, through the nanoscale-cavity effect, either the energy transfer from the donor into acceptor or the acceptor emission efficiency can be enhanced, leading to an improved color conversion process.

A surface plasmon (SP) resonance on a metal nanostructure can couple with a light emitter to enhance its emission efficiency and to produce strong electromagnetic field distributions around the metal nanostructure and light emitter [7,8]. If the acceptor in a FRET process is placed within such a field distribution produced by the SP-coupled donor, the energy absorbed by the acceptor or the FRET efficiency can be increased [9,10]. However, the SP coupling can also enhance donor emission that reduces the available energy for transferring into the acceptor. Therefore, whether SP coupling can enhance the efficiency of an FRET process depends on the SP coupling condition [11]. If the SP coupling condition can be properly designed, both FRET efficiency and acceptor emission efficiency can be enhanced through SP coupling, leading to an effective color conversion process. In particular, by applying SP coupling to a color conversion system with a nanoscale cavity, the color conversion efficiency can be further enhanced.

In this paper, we studied the behaviors of the color conversion from a blue-emitting InGaN/GaN quantum-well (QW) structure into the emission of a colloidal quantum dot (QD), which is inserted into a GaN porous structure (PS) near the QW structure [12,13,14], under the condition of SP coupling. Both green-emitting QD (GQD) and red-emitting QDs (RQD) were used in this study. The SP coupling is induced by depositing Ag nanoparticles (NPs) onto the top surface of a sample. The color conversion behaviors were investigated with the measurements of continuous-wave (CW) and time-resolved (TR) photoluminescence (PL) spectroscopies. The results of a simulation study well support the experimentally observed enhancement of color conversion through the SP coupling. In Section 2 of this paper, the sample structures, their fabrication procedures, and the optical measurement methods are described. The optical measurement results are presented in Section 3. The simulation study is reported in Section 4. Further discussions about the results are made in Section 5. Finally, conclusions are drawn in Section 6.

## 2. Sample Structures, Fabrication Procedures, and Measurement Methods

Figure 1a schematically illustrates the sample structure under study, in which a PS layer of 200 nm in thickness was fabricated in the grown highly Si-doped GaN (n^+^-GaN) layer. On the top surface, Ag NPs are deposited for inducing the SP coupling with the QW structure. To prepare such a QW template, first a 200-nm n^+^-GaN layer with the Si doping concentration at ~2 × 10^19^ cm^−3^ was grown on a ~3-μm un-doped GaN (u-GaN) layer, followed by the deposition of a ~50 nm n-GaN layer with the Si doping concentration at ~5 × 10^18^ cm^−3^. The growth temperatures for the n^+^-GaN, n-GaN, and u-GaN layers are the same at 1040 °C. Then, a 5-period InGaN/GaN QW structure was grown at the temperatures of 695 and 793 °C for the InGaN well layers and GaN barrier layers, respectively. The QW structure was capped by a u-GaN layer of ~30 nm in thickness. At room temperature, the emission peak wavelength of the QW structure lies between 450 and 460 nm. To blue-shift the localized surface plasmon (LSP) resonance feature of the surface Ag NPs such that it is close to the QW emission peak wavelength, a ~5-nm SiO_2_ layer was coated onto the top surface before Ag NP deposition [15,16]. GQDs and RQDs were inserted into the nanoscale cavities of the PS layer for studying their emission behaviors. CdZnSeS/ZnS GQDs and RQDs were purchased from Taiwan Nanocrystals Inc. Hsinchu, Taiwan. They are capped with an amphiphilic polymer, i.e., poly(isobutylene-alt-maleic anhydride), and hence are negatively charged with zeta potentials between −40 and −50 mV [17]. Including the capped amphiphilic polymer, the size of a GQD or RQD ranges from 8 through 10 nm. The emission wavelengths of GQD and RQD are ~530 and ~625 nm, respectively.

Five samples were prepared for comparing their emission performances, including samples A and P without QD, and samples G, R, and G+R with designated QDs. Without a PS in sample A, surface Ag NPs were deposited for observing the SP coupling effect on QW emission. With a PS and surface Ag NPs in sample P, the results show the SP coupling effect on the QW emission when a nearby PS exists. In samples G, R, and G+R, GQD only, RQD only, and GQD plus RQD, respectively, were inserted into the individual PSs for studying the SP coupling effects of surface Ag NPs on the FRETs from QW into QD and QD emission. The electrochemical etching conditions for fabricating a PS include 15 V in applied voltage, 5 wt% for the HNO_3_ electrolyte concentration, and 5.5 min for the etching duration. In the etching process, the sample was electrically connected with an indium ball as the anode and a Pt plate was used as the cathode. Figure 1b shows the cross-sectional scanning electron microscopy (SEM) image of the PS in sample P. We can see that the whole n^+^-GaN layer has been etched to become a PS. The cross-sectional facet in this image is perpendicular to the direction of current injection in the electrochemical process. On the sample top surface, the direction along (perpendicular to) this facet is designated as the perpendicular, i.e., ⊥, (parallel, i.e., //,) polarization in optical measurements because the fabricated nano-pores essentially extend along the direction perpendicular to this facet. Basically, the electrochemical etching path follows the flow direction of injected electric current. Therefore, the pore extension essentially follows the injected current direction. However, because it is difficult to well control the current flow path inside a sample, the PS anisotropy in the lateral dimension is quite weak. Nevertheless, as shown in the following discussions, we can still clearly observe the anisotropic optical properties in a sample with QDs inserted into a PS and/or Ag NPs on the top surface due to the anisotropic nanoscale-cavity effect. It is noted that although the LSP resonance of surface Ag NPs is polarization-independent, the SP coupling is polarization-dependent due to the anisotropic PS in a sample, which results in a polarization-dependent nanoscale-cavity effect.

To insert QDs into a PS, we fabricated a linear array of surface hole of 300 μm in diameter and ~450 nm in depth. After a droplet of the ethanol solution of QD was applied to the sample surface, the solution could flow into the PS through the surface holes. For spreading the QD solution laterally in the PS, the sample was spun at 300 rpm in speed for 30 min. To focus the study on the emission behaviors of the QDs inserted into the subsurface PS, we cleaned the sample’s surface with wet cotton swabs to remove the QDs remained on the surface. The sample’s surface was then examined with SEM to make sure that it was indeed clean. Figure 1c shows the cross-sectional dark-field transmission electron microscopy (TEM) image of sample R. In this image, above the PS, we can see five bright stripes, which correspond to the five QWs. On the top surface, Ag NPs can be observed. In the PS, the foggy regions correspond to the distributions of QD. Due to the surface coating of the amphiphilic polymer, the particle-like image of a QD becomes blurred in the TEM image. However, the scanning image of energy-dispersive X-ray spectroscopy (EDX) in Figure 1d can prove the distribution of QD in the PS. In this image, the yellow, green, red, and pink dots show the distributions of elements Zn, Cd, Se, and S, respectively.

The surface Ag NPs were fabricated by depositing Ag of 1.8 nm in thickness onto the SiO_2_-coated sample surface at room temperature. Without a thermal annealing process, the deposited Ag naturally formed an NP distribution on the sample surface, as illustrated with the SEM image (sample P) in the inset of Figure 2. The three surface nano-holes in this SEM image were formed during the electrochemical etching process. These surface nano-holes help in directing the electrolyte into the n^+^-GaN layer and letting out the gases (N_2_ and O_2_) produced during the electrochemical etching process. The main body of Figure 2 shows the normalized transmission spectra of the five samples under study. They were obtained by using the individual transmission spectra before Ag NP deposition as the normalization baselines. Here, the transmission depressions correspond to the LSP resonance features of the surface Ag NPs with the resonance peaks all around 465 nm. In Figure 2, the three vertical dashed lines indicate the emission peak wavelengths of the QW structure, GQD, and RQD. We can see that the LSP resonance peaks are close to the QW emission peak wavelength for producing strong SP coupling with the QW structure. At the GQD emission wavelength, the LSP resonance is still quite strong. However, because the distance between the surface Ag NPs and the GQD distribution in the PS is quite large (>160 nm), their SP coupling is expected to be weak. The fabrication of samples G, R, or G+R followed the procedure of PS fabrication, QD insertion, and then Ag NP deposition.

## 3. Optical Characterization Results

The CWPL measurement was excited by an InGaN laser diode of 405 nm in wavelength and 6 mW in output power. The TRPL measurement was excited by the second-harmonic (390 nm in wavelength and ~1.5 mW in power) of a femtosecond Ti:sapphire laser (MIRA 900, pumped by VERDI-8W, Coherent, USA) with the pulse repetition rate at 76 MHz. The signals were monitored with a photon-counting system (the time-correlated single photon-counting solution delivered by Becker & Hickl). The temporal resolution of this system was higher than 0.1 ns. The method for calibrating the decay time of a PL decay profile has been reported in a previous publication [18]. Figure 3a shows the normalized CWPL spectra of sample A at 10 and 300 K before (labeled by “intrinsic” or “I”) and after (labeled by “Ag NP” or “NP”) surface Ag NP deposition. The internal quantum efficiency (IQE) of a QW structure is defined as the ratio of the integrated intensity at 300 K over that at a low temperature, i.e., 10 K in this study. The IQEs for sample A before and after Ag NP deposition are also shown in Figure 3a. The IQE is increased from the intrinsic value of 57.8% to 63.8% after Ag NP deposition. The SP coupling indeed enhances the IQE by a significant amount. Figure 3b shows that the results of sample P are similar to those in Figure 3a for sample A. However, one more set of data obtained at the stage after PS fabrication but before Ag NP deposition (labeled by “PS”) was added. Here, we can see that after PS fabrication, the PL spectral peaks are blue-shifted at either temperature. As also shown in the figure, the IQE of this sample increases from the intrinsic value of 57.4% to 67.2% after PS fabrication, and then to 69.4% after Ag NP deposition. The IQE increase after PS fabrication is caused by the strain relaxation and hence the weakening of the quantum-confined Stark effect (QCSE) in the QW structure [19]. The weakened QCSE leads to the increase of radiative recombination or IQE and the blue-shift of the emission spectrum. Figure 3c,d show the results of samples G and R, respectively, similar to Figure 3a,b. We can again see that after PS fabrication, the IQE is increased and the emission spectrum is blue-shifted. It is noted that the evaluation for QW IQE based on the PL measurements at 10 and 300 K becomes unreliable after QDs are inserted into the PS of a sample. This is because the inserted QDs absorb the QW emission and distort the measurement of its intensity. The behaviors of the normalized CWPL spectra in sample G+R are similar to those in either sample G or R and are not shown in this paper. Nevertheless, the variation of its IQE will be given later.

Figure 4 shows the normalized blue-light PL decay profiles of samples A and P at different fabrication stages. After PS fabrication in sample P, the PL decay rate significantly increased, which is consistent with the increase of IQE. After Ag NP deposition (labeled by “SP”) in sample A, the PL decay rate is also increased that is again consistent with the increase of IQE after SP coupling was introduced. In sample P, after Ag NP deposition, the PL decay rate in either polarization is also enhanced. Although the decay profiles in the two polarizations are close to each other, the decay rate in the ⊥-polarization is slightly higher. Figure 5 shows the normalized blue-light PL decay profiles of sample G+R at different fabrication stages. Here, we can see that after PS fabrication, the decay rate is increased. After QD insertion but before Ag NP deposition (w/o SP), the QW PL decay rate is further increased in either polarization due to the FRET from QW into QD. The PL decay profiles in the two polarizations almost coincide with each other. Then, after Ag NP deposition (SP), the PL decay rate is further increased in either polarization due to SP coupling. With SP coupling, the PL decay profiles in the two polarizations become clearly separated. The decay rate in the ⊥-polarization is higher.

Figure 6 and Figure 7 show the normalized green- (red-) light PL decay profiles of samples G (R) and G+R in the cases with and without SP coupling for the two polarizations. For comparison, that of GQD (RQD) placed on the top surface of a GaN template, which does not contain a QW structure, is also shown, as labeled by “GQD (intrinsic)” (“RQD (intrinsic)”). Compared with the intrinsic GQD decay profile, the green-light decay rate in either polarization is decreased when GQD is inserted into the PS of sample G. This decrease of decay rate is caused by the FRET from QW into GQD. After SP coupling is introduced, the green-light decay rate is further decreased in sample G, indicating that the SP coupling can enhance the FRET from QW into GQD. Although the decay profiles for the two polarizations are close to each other in either case with or without SP coupling, we can still see the higher decay rate in the ⊥-polarization. In sample G+R, generally the green-light decay rates are significantly increased, when compared with those in sample G, due to the FRET from GQD into RQD in the PS. In this sample, SP coupling can also enhance the FRET from QW into GQD such that the green-light decay rate is slightly reduced. Here, the green-light decay profiles in the two polarizations are clearly separated. As shown in Figure 7, the red-light decay rates in sample R are also reduced, when compared with the intrinsic case, due to the FRET from QW into RQD. In this sample, the SP coupling effect for enhancing the FRET from QW into RQD is weak. Additionally, the difference between the two polarizations is small. Then, in sample G+R, the red-light decay rate is further reduced because RQD receives energy from GQD through the FRET from GQD into RQD. In this sample, the SP coupling can further reduce the red-light decay rate. In either case, with or without SP coupling, again, the difference between the two polarizations is small.

In Table 1, we show the decay times of the QW structure at different fabrication stages in the five samples under study. The numbers inside the curly brackets show the corresponding IQE values. The numbers inside the parentheses show the FRET efficiencies in the corresponding FRET processes. The FRET efficiency, η, is defined as η = 1-τ_DA_/τ_D_, where τ_DA_ (τ_D_) is the PL decay time of the energy donor when the acceptor is present (absent) [11]. Here, in each sample, the QW PL decay time decreases after PS fabrication, after QD insertion, or after Ag NP deposition. The decay time in the ⊥-polarization is always shorter than that in the //-polarization. A shorter decay time always corresponds to a higher IQE. In sample G, the FRET efficiency increases from 11.92 (12.91%) at the stage after QD insertion, but before Ag NP deposition, to 19.70 (23.51%) after Ag NP deposition in the //-(⊥-) polarization. In sample R, the FRET efficiency increases from 17.89 (19.06%) at the stage after QD insertion but before Ag NP deposition to 22.41 (24.92%) after Ag NP deposition in the //- (⊥-) polarization. Then, in sample G+R, the FRET efficiency increases from 37.50 (37.67%) at the stage after QD insertion but before Ag NP deposition to 60.33 (64.67%) after Ag NP deposition in the //-(⊥-) polarization. We can see that SP coupling can indeed enhance FRET efficiency. In particular, the FRET efficiency enhancement through SP coupling is stronger in the ⊥-polarization. The FRET efficiencies in sample R are generally higher than those in sample G. However, the SP coupling effect for enhancing FRET efficiency is weaker in sample R, when compared with sample G. With both FRETs from QW into GQD and RQD, the overall FRET efficiencies in sample G+R are higher than those in the other two QD samples.

In Rows 3 and 4 of Table 2, we show the green- and red-light decay times, respectively, in the two polarizations under the conditions with and without SP coupling for the three QD samples. For comparison, the intrinsic PL decay times of GQD and RQD, i.e., 5.78 and 8.95 ns, respectively, are shown inside the curly brackets in Row 1. For the green light in the //-(⊥-) polarization, through the FRET from QW into GQD, the decay time increases from the intrinsic value of 5.78 ns to 6.26 (6.22) ns in the case without SP coupling, and then to 6.57 (6.48) ns under the condition of SP coupling in sample G. For the red light in the //-(⊥-) polarization, through the FRET from QW into RQD, the decay time increases from the intrinsic value of 8.95 ns to 9.27 (9.23) ns in the case without SP coupling, and then to 9.43 (9.39) ns under the condition of SP coupling in sample R. In sample G+R, for the green light in the //-(⊥-) polarization, the decay time decreases to 3.73 (3.69) ns in the case without SP coupling through the FRET from GQD into RQD, but returns to 3.98 (3.80) ns under the condition of SP coupling. On the other hand, for the red light in the //-(⊥-) polarization in this sample, the decay time increases to 11.15 (11.09) ns in the case without SP coupling, and further increases to 11.87 (11.84) ns under the condition of SP coupling.

Figure 8a–c show the normalized CWPL spectra in the two polarizations under the conditions with and without SP coupling for samples G, R, and G+R, respectively. In each sample, under either condition with or without SP coupling, the spectra of the two polarizations are normalized with respect to the blue intensity peak in the //-polarization. Here, one can see that in either the green- or red-light component, the intensity in the ⊥-polarization is always higher than that in the //-polarization. However, the blue-light intensity in the ⊥-polarization is always lower than that in the //-polarization. These results clearly indicate that the FRET from QW into QD in the ⊥-polarization is stronger. The results in Figure 8 also show that the SP coupling can indeed enhance the green- and red-light intensity ratios with respect to the blue-light intensity. In Rows 5 and 6 of Table 2, we show the ratios of the integrated intensities of green and red lights over those of the corresponding blue lights, i.e., the G/B and R/B ratios, respectively, in the two polarizations under the conditions with and without SP coupling. The blue-, green-, and red-light components in a spectrum are separated by dividing the spectrum at 510 and 580 nm in wavelength. Under either condition with or without SP coupling, either the G/B or R/B ratio in the ⊥-polarization is always larger than the corresponding value in the //-polarization for each sample. By introducing SP coupling to a sample, either the G/B or R/B ratio in each polarization is increased. In the last three rows of Table 2, we show the polarization ratios of the three samples under the conditions with and without SP coupling. A polarization ratio is defined as the ratio of the integrated intensity in the ⊥-polarization over that in the //-polarization. We can see that all the green- and red-light (blue-light) polarization ratios are larger (smaller) than unity. In each sample, the SP coupling can reduce the blue-light polarization ratio because it enhances the FRET more effectively from QW into QD in the ⊥-polarization, when compared with the //-polarization. The SP-coupling effect also reduces the polarization ratios of either green or red light in samples G and G+R.

## 4. Simulation Study

To further understand the mechanism of color conversion enhancement in the sample structure, a numerical simulation study was undertaken. The simulation structures are schematically illustrated in Figure 9a–c. In Figure 9a for structure S with SP coupling, an infinitely-long empty nano-tube with the radius at r_0_ = 30 nm was horizontally embedded in a QW template. The QW is represented by a dipole, denoted by the blue arrow, which serves as the donor in the FRET to be studied. A QD was placed inside the nano-tube and serves as the acceptor of the FRET from QW into QD. A 5-nm SiO_2_ layer (refractive index at 1.5) was placed between the surface Ag NP and the QW template. It is assumed that the center of the Ag NP, the QW-dipole donor, and the QD-acceptor are vertically aligned. All of them lie in a vertical plane passing the axis of the embedded nano-tube. The distance between the QW-dipole donor and the top surface of the QW template (the upper boundary of the nano-tube) is d = 60 nm (s = 60 nm). The distance between the QD-acceptor and the upper boundary of the nano-tube is t = 10 nm. As illustrated in Figure 9b, the surface Ag NP is a truncated ellipsoid in geometry with the horizontal semi-axis of b = 21.8 nm, the height of h = 30 nm, and the radius of the circular contact interface of 18.9 nm. For comparison, in the reference structure (structure SR), as illustrated in Figure 9c, the Ag NP and SiO_2_ layer are removed to show the results without SP coupling. The numerical simulation method has been described in a few earlier publications [7,16]. In numerical computations, the refractive index of GaN is set at 2.399. Additionally, the experimental data were used for the wavelength-dependent dielectric constant of Ag [20]. To evaluate the radiation behavior of a dipole, we first computed its radiated electromagnetic field when it was placed in a homogeneous spherical background space. Then, the total field was evaluated in the real structure for simulation based on the commercial software of COMSOL. By subtracting the radiated field of the dipole from the total field, we obtained the scattered field, which was used for evaluating the feedback effect on the dipole radiation behavior from the surrounding structure. With the obtained scattered field, the optical Bloch equations were solved to give the strength and orientation of the dipole modified by the feedback effect. Based on the modified dipole, the final electromagnetic field distribution and the total radiated power can be computed. With the feedback process, the effect of the scattered field caused by the nano-tube on the radiation behavior of the dipole was included. In other words, the Purcell effect was practically taken into account [6].

Based on the simulation study, Figure 10a,b show the y-z-plane distributions of electric field strength (norm) at 455 nm in structures S and SR, respectively, produced by an x-oriented donor dipole. This radiating dipole is located at the center (marked by “x”) of the white circular region, in which no field distribution is shown because of the extremely strong field there. Here, we can see that inside the nano-tube, the electromagnetic field is significantly stronger in structure S, when compared with structure SR, even though Figure 10a,b is plotted in the log scale, indicating the SP-coupling effect induced by the surface Ag NP.

Figure 11a shows the normalized field intensity produced by the QW-dipole donor at the position of the QD acceptor under the conditions with (structure S) and without (structure SR) SP coupling in the cases of x- and y-dipole. The results here are normalized with respect to the corresponding field intensities in a structure without the Ag NP, SiO_2_ layer, and the embedded nano-tube, i.e., an air/GaN half-space structure. For either the x- or y-dipole in structure S, we can see an intensity peak around 460 nm, which is caused by the SP-coupling effect, confirming that the SP coupling can indeed enhance the donor field intensity inside the nano-tube. Without SP coupling in structure SR, the donor intensity inside the nano-tube can also be enhanced due to the nanoscale-cavity effect for the x-dipole, which corresponds to the ⊥-polarization in the experimental study reported earlier in this paper. The vertical dashed line in Figure 11a indicates the experimental QW emission peak wavelength at 455 nm. Figure 11b shows the normalized radiated powers of the QD-acceptor in structures S and SR for the x- and y-dipole. The results are normalized with respect to the radiated power in a homogeneous space of air. Here, except for the depressions around 460 nm, the results in structures S and SR are close to each other for either dipole orientation. The depressions are caused by the SP coupling between the QD-acceptor and the LSP resonance of the surface Ag NP. In other words, based on our simulation model, with the distance between the QD-acceptor and the bottom face of the Ag NP at 135 nm, the SP coupling leads to an emission suppression around the LSP resonance peak wavelength of the surface Ag NP. However, this behavior is unimportant in our numerical study because we are mainly concerned with the acceptor emissions in the green and red spectral ranges, particularly at 530 and 625 nm in wavelength, as indicated by the vertical dashed lines in Figure 11b. The larger-than-unity results shown in Figure 11b indicate that the emission efficiency of a QD inside a nanoscale cavity can indeed be enhanced, particularly in the polarization perpendicular to the nano-tube axis (the x-dipole). Rows 2 and 3 of Table 3 show the normalized donor intensities at the position of the acceptor in structures S and SR, respectively, for the x- and y-dipole at the wavelength of 455 nm. The donor intensities are stronger for the x-dipole. Rows 4 and 5 (6 and 7) of Table 3 show the normalized radiated powers of the acceptor in structures S and SR, respectively, for the x- and y-dipole at the wavelength of 540 (625) nm. The dipole polarization dependencies can also be observed.

Figure 12 shows the ratios of the intensity produced by the donor at the position of the acceptor and the radiated power of the acceptor in structure S over the corresponding values in structure SR. These ratios show the enhancements of those two parameters through SP coupling. We can see that although the SP coupling produces weak effects on the emissions of GQD and RQD because their emission wavelengths are far away from the LSP resonance peak, it can effectively increase the donor field intensity inside the nano-tube. In Rows 8–10 of Table 3, we show the donor intensity ratios at 455 nm and the acceptor radiated power ratios at 530 and 625 nm, respectively, for the x- and y-dipole. The multiplication of the donor intensity ratio at 455 nm by the acceptor radiated power ratio at 530 (625) nm corresponds to the enhancement ratio of the color conversion efficiency from QW into GQD (RQD), as shown in Row 11 (12) of Table 3. Although the SP coupling does not produce a significant change of the emission efficiency of a QD inside the nanoscale cavity, it does enhance the donor intensity at the position of the QD, i.e., the FRET from QW into QD, resulting in a significant enhancement of the color conversion efficiency.

## 5. Discussions

As shown in Row 7 of Table 2, the SP coupling leads to the decrease of blue-light polarization ratio in each sample due to the stronger SP-coupling enhancement of the FRET from QW into QD in the ⊥-polarization. In this situation, we may expect the larger polarization ratios for green and red lights because the GQD and RQD receive more energy from the QW structure through FRET in the ⊥-polarization. However, as shown in Rows 8 and 9 of Table 2, after the SP coupling effect is introduced, the green- and red-light polarization ratios are decreased except for that in sample R, even though they are still larger than unity. This result may have two attributions. First, in the CWPL measurement, the obtained PL intensities include the contribution of the far-field process of photon emission–absorption–reemission, besides that of the near-field process of FRET. The far-field process is more weakly polarization-dependent, when compared with the near-field process. Second, the non-resonant scattering of the surface Ag NP distribution can also weaken the polarization dependence of the measured CWPL intensity for green or red light. It is noted that in our simulation model we did not really evaluate the transferred power from the donor into the acceptor and hence we could not observe such an experimental result in the simulation study.

In Figure 8, we can see that certain spectral peaks are shifted after the SP coupling effects are introduced to the samples. Those of blue light in all the three samples are blue-shifted after the SP coupling effects are included. This is because the SP coupling and hence the FRET from QW into QD are stronger on the long-wavelength side of the QW emission spectrum, as shown in Figure 2, such that the QW emission on the long-wavelength side becomes weaker. Therefore, the QW emission peak is effectively blue-shifted. The reason for the noticeable red-shift of the green-light peak in sample G is complicated. It can be caused by the non-uniform GQD emission wavelength. It can also be attributed to the wavelength-dependent nanoscale-cavity and SP-coupling effects.

## 6. Conclusions

In summary, we have designed and implemented a device structure for improving the color conversion performance from a QW structure into QDs inserted into a nearby subsurface GaN PS by introducing the SP coupling between the QW structure and the LSP resonance on surface Ag NPs to the device. Based on TRPL and CWPL measurements, we observed the enhanced FRET efficiencies from QW into QD through SP coupling. The SP-coupling induced enhancement was stronger in the polarization perpendicular to the essentially extended direction of the pores in the fabricated PS. The results of a numerical simulation study well supported the experimental observations. The encouraging results in this paper can help us in improving the color conversion efficiency in a light-emitting device.

## Figures and Tables

**Figure 1 nanomaterials-13-00328-f001:**
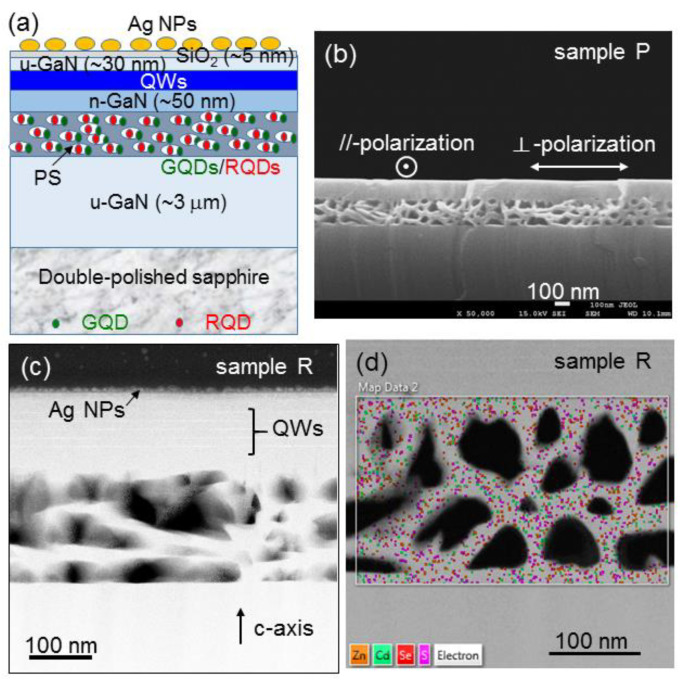
(**a**) Schematic illustration for the sample structure. The PS was fabricated in the grown n^+^-GaN layer for accommodating GQD and/or RQD. Surface Ag NPs were deposited to induce the SP coupling with the QW structure. (**b**) Cross-sectional SEM image showing the PS in sample P. The directions of the two polarizations are defined in this figure. (**c**) Cross-sectional TEM image of sample R. (**d**) EDX mapping result in a TEM image of sample R showing the distributions of the four composition elements of RQD.

**Figure 2 nanomaterials-13-00328-f002:**
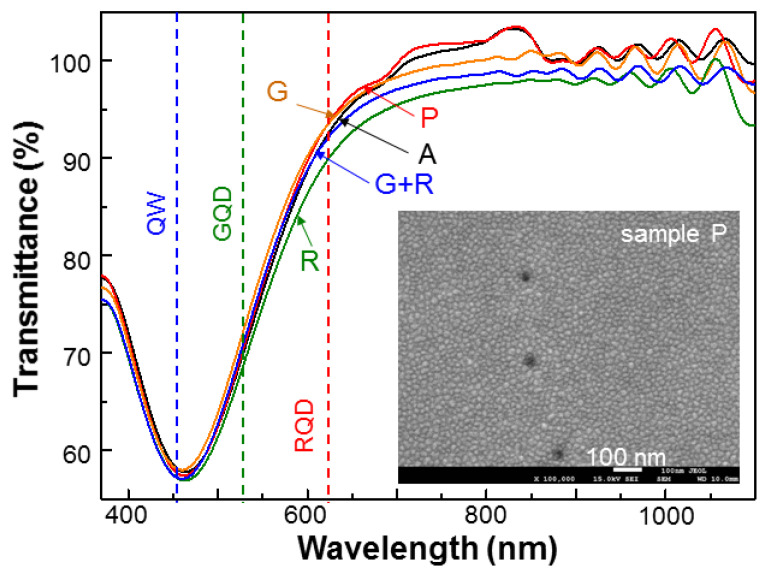
Normalized transmission spectra of the five samples under study after surface Ag NPs were deposited. The three vertical dashed lines indicate the emission peak wavelengths of the QW structure, GQD, and RQD. The inset shows the SEM image of sample P after surface Ag NPs were deposited.

**Figure 3 nanomaterials-13-00328-f003:**
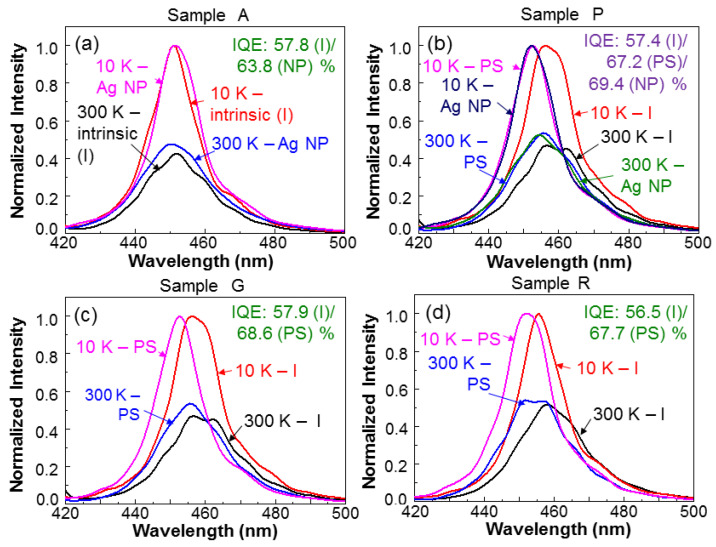
(**a**–**d**): Normalized QW CWPL spectra in different fabrication stages at 10 and 300 K for samples A, P, G, and R, respectively. I: intrinsic condition; PS: after PS fabrication; NP: after surface Ag NP deposition.

**Figure 4 nanomaterials-13-00328-f004:**
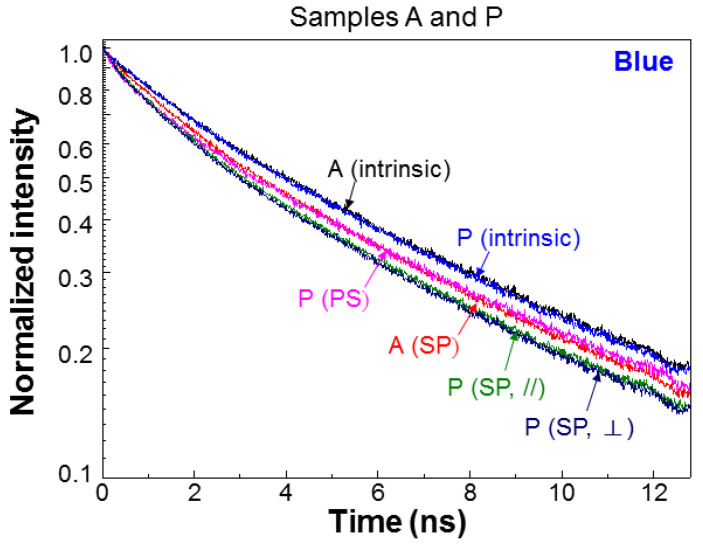
Normalized blue-light PL decay profiles of samples A and P at different fabrication stages. PS: after PS fabrication; SP: after surface Ag NP deposition for inducing SP coupling.

**Figure 5 nanomaterials-13-00328-f005:**
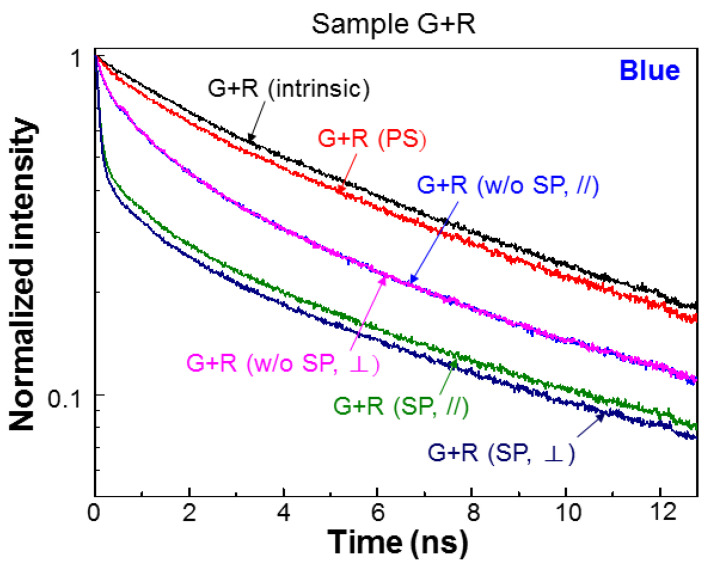
Normalized blue-light PL decay profiles of sample G+R at different fabrication stages in the two polarizations. PS: after PS fabrication; w/o SP: after QD insertion but before Ag NP deposition; SP: after QD insertion and Ag NP deposition.

**Figure 6 nanomaterials-13-00328-f006:**
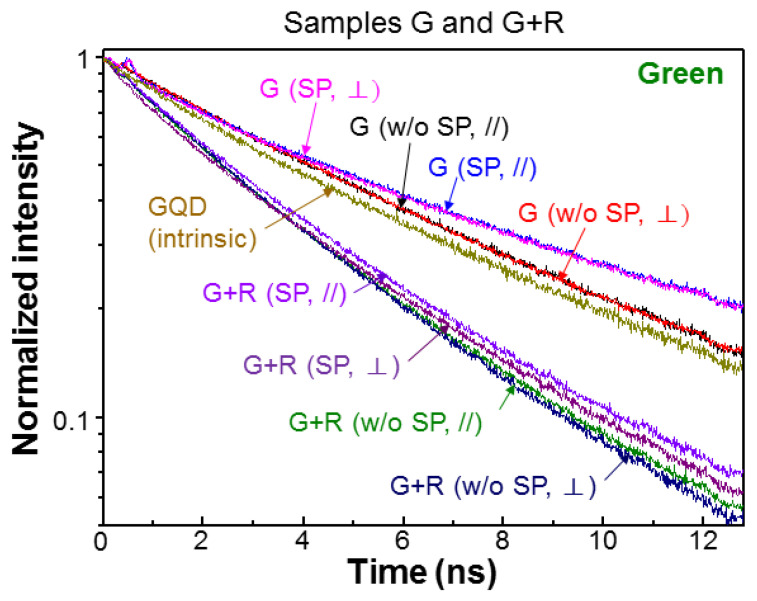
Normalized green-light PL decay profiles of samples G and G+R at different fabrication stages in the two polarizations.

**Figure 7 nanomaterials-13-00328-f007:**
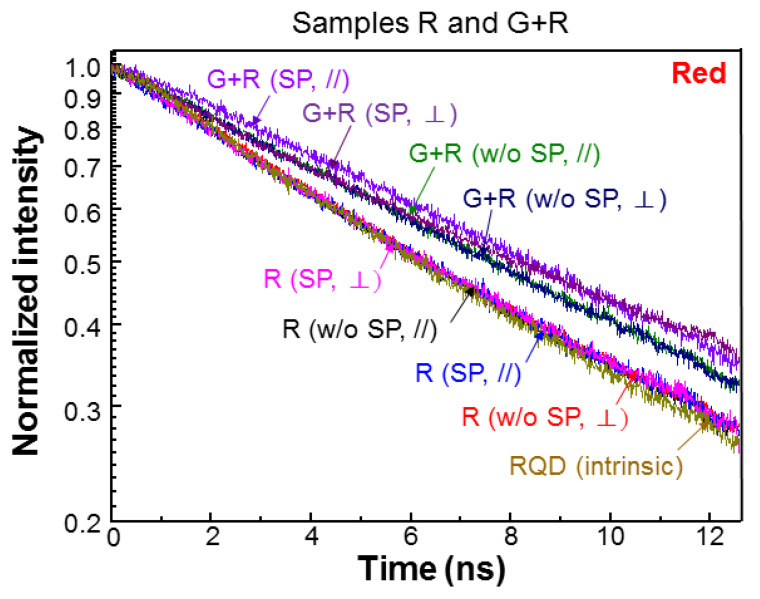
Normalized red-light PL decay profiles of samples R and G+R at different fabrication stages in the two polarizations.

**Figure 8 nanomaterials-13-00328-f008:**
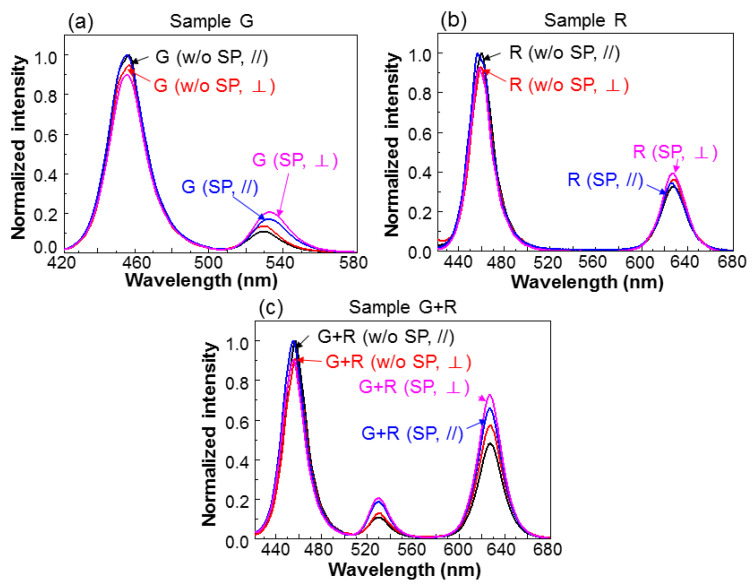
(**a**–**c**): Normalized CWPL spectra before (w/o SP) and after (SP) Ag NP deposition in the two polarizations for samples G, R, and G+R, respectively.

**Figure 9 nanomaterials-13-00328-f009:**
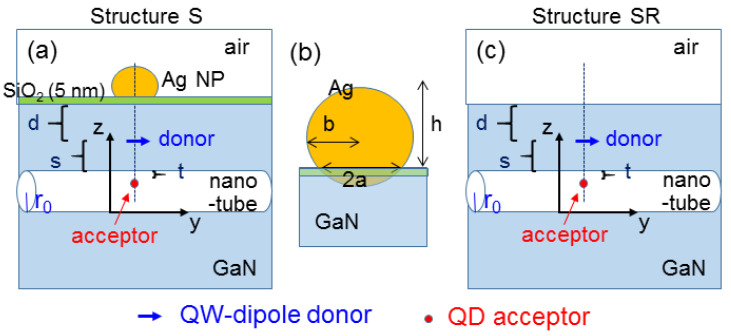
(**a**): Illustration of structure S used in the simulation, including a surface Ag NP for inducing the SP coupling with the QW-dipole donor and a QD-acceptor inside an embedded nano-tube. (**b**) Geometry illustration of the surface Ag NP. (**c**): Illustration of the reference structure SR for simulation.

**Figure 10 nanomaterials-13-00328-f010:**
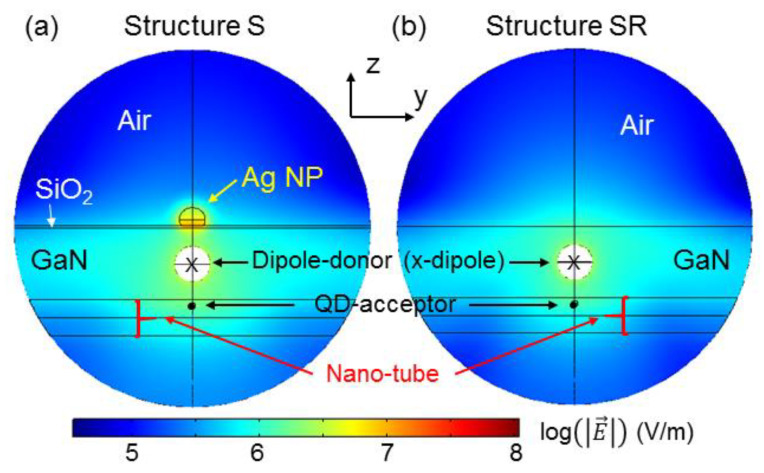
(**a**,**b**): y-z-plane distributions of electric field strength (norm) at 455 nm in structures S and SR, respectively, produced by an x-oriented donor dipole.

**Figure 11 nanomaterials-13-00328-f011:**
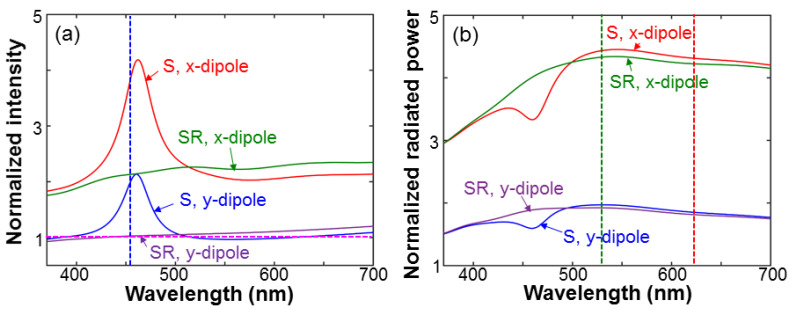
(**a**): Normalized intensity spectra produced by the QW-dipole donor at the position of the QD-acceptor in structures S and SR for the two dipole orientations. (**b**): Normalized radiated power spectra of the QD-acceptor in structures S and SR for the two dipole orientations.

**Figure 12 nanomaterials-13-00328-f012:**
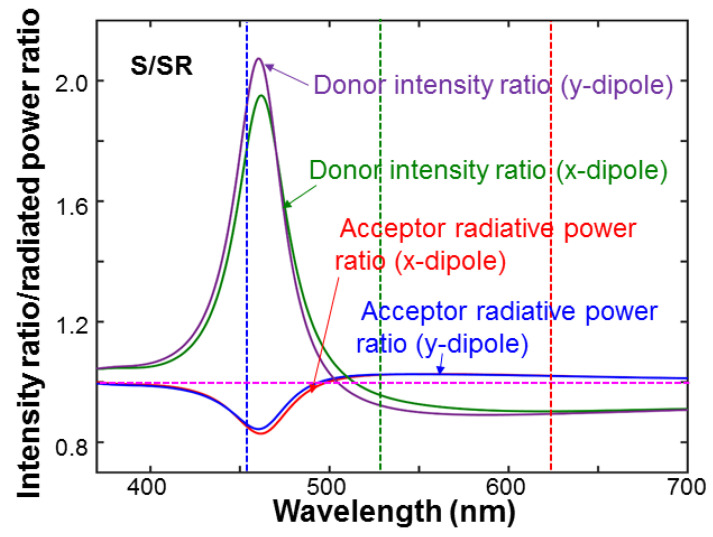
Ratios of the donor intensity and acceptor radiated power in structure S over the corresponding values in structure SR for the two dipole orientations.

**Table 1 nanomaterials-13-00328-t001:** Blue-light PL decay times (in ns) in the five samples under study at different fabrication stages. The numbers inside the curly brackets show the corresponding IQEs. The numbers inside the parentheses show the corresponding FRET efficiencies.

Sample		A	P	G	R	G+R
Intrinsic		6.48 {57.8%}	6.36 {57.4%}	6.65 {57.9%}	6.67 {56.5%}	6.46 {57.7%}
After PS fabrication		---	5.89 {67.2%}	6.04 {68.6%}	5.98 {67.7%}	6.00 {68.1%}
After QD insertion	//	---	---	5.32 (11.92%)	4.91 (17.89%)	3.75 (37.50%)
	⊥	---	---	5.26 (12.91%)	4.84 (19.06%)	3.74 (37.67%)
After Ag NP deposition	//	5.71 {63.8%}	5.33 {69.4%}	4.85 (19.70%)	4.64 (22.41%)	2.38 (60.33%)
	⊥		5.23	4.62 (23.51%)	4.49 (24.92%)	2.12 (64.67%)

**Table 2 nanomaterials-13-00328-t002:** Green- and red-light PL decay times for samples G, R, and G+R in the two polarizations before (w/o SP) and after (SP) Ag NP deposition. The intrinsic PL decay times of GQD and RQD are shown inside the curly brackets in row 1. Also shown in this table are the color ratios, i.e., G/B and R/B ratios, and the polarization ratios (PRs) for the three color components.

Sample	G (//, ⊥) {5.78 ns}		R (//, ⊥) {8.95 ns}		G+R (//, ⊥)	
SP condition	w/o SP	SP	w/o SP	SP	w/o SP	SP
Green decay time (ns)	6.26, 6.22	6.57, 6.48	---	---	3.73, 3.69	3.98, 3.80
Red decay time (ns)	---	---	9.27, 9.23	9.43, 9.39	11.15, 11.09	11.87, 11.84
G/B ratio	0.107, 0.137	0.172, 0.225	---	---	0.114, 0.146	0.174, 0.213
R/B ratio	---	---	0.365, 0.412	0.371, 0.447	0.530, 0.694	0.696, 0.850
Blue PR	0.948	0.901	0.933	0.897	0.914	0.902
Green PR	1.217	1.179	---	---	1.169	1.103
Red PR	---	---	1.084	1.090	1.197	1.102

**Table 3 nanomaterials-13-00328-t003:** Simulation results obtained from Figure 11 and Figure 12 at 455, 530, and 625 nm in wavelength for the two dipole orientations, including the results of donor intensity, acceptor radiated power, and color conversion enhancement factor.

	x-Dipole	y-Dipole
Normalized donor intensity at 455 nm (S)	3.872	2.016
Normalized donor intensity at 455 nm (SR)	2.132	1.027
Normalized acceptor radiated power at 530 nm (S)	4.435	1.970
Normalized acceptor radiated power at 530 nm (SR)	4.331	1.922
Normalized acceptor radiated power at 625 nm (S)	4.309	1.847
Normalized acceptor radiated power at 625 nm (SR)	4.222	1.811
Donor intensity ratio at 455 nm (S/SR)	1.816	1.963
Acceptor radiative power ratio at 530 nm (S/SR)	1.024	1.025
Acceptor radiative power ratio at 625 nm (S/SR)	1.021	1.020
Color conversion enhancement factor from 455 into 530 nm (S/SR)	1.860	2.012
Color conversion enhancement factor from 455 into 625 nm (S/SR)	1.854	2.002

## Data Availability

All the data supporting reported results can be found in the text of this paper.

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
