# Peer review of "Effects of Surface Plasmon Coupling on the Color Conversion of an InGaN/GaN Quantum-Well Structure into Colloidal Quantum Dots Inserted into a Nearby Porous Structure"

_nanomaterials, 2023, doi:10.3390/nano13020328_

Round 1
Reviewer 1 Report
The manuscript by S. Yang et al. demonstrates the characteristics of the color conversion from the blue-emitting quantum well (QW) structure into the emission of green- and red-emitting quantum dot (QD) inserted into the porous structure (PS), where the surface plasmon (SP) resonance coupling is induced by depositing Ag nanoparticles (NPs) on the surface of the structure. The manuscript is supported by a nice introduction, and investigations to support their assertions and conclusions. The manuscript could appeal to the related audiences. To be published the following issues be addressed.
Suggestions for revision:
1) The authors describe that the valley in the transmission spectra shown in Fig. 2 is attributed to the localized surface plasmon resonance (LSPR). It would be informative if the authors can show the transmission spectrum of the sample without the deposition of the Ag NPs or the electric field profiles at 465 nm to validate that the LSPR is excited at the wavelength of 465 nm.
2) After the deposition of the Ag NPs, a shape of the Ag NPs would be close to a hemi-sphere, which would present the polarization-insensitive performance. The authors state that the decay rate in the vertical polarization is higher that in the parallel polarization in Figs. 4, 5, 6, and 7. What is the reason for this?
3) The shorter decay time corresponds to the higher internal quantum efficiency (IQE). It would be instructive if the authors can exhibit the measured IQE spectra of the sample P, GQD, RQD and GQD + RQD.
4) It would be informative if the authors can provide more details on the simulation method to plot Fig. 10, possibly with the related equations.
5) Although the manuscript is overall well-written, there are many sentences with missing articles, which should be revised.
Reviewer 2 Report
[1] The abbreviations and technical terms are not defined appropriately. Abbreviations and acronyms should be typically defined the first time the term is used within the manuscript and then used throughout the remainder of the manuscript. Additionally, GQD and RQD are used in two meanings: QD itself and structures including QDs.
[2] As the authors describe in the introduction, the spectral and spatial overlap between electric-field and dipole of the emitter is critical to understand the FRET enhancement mechanism. Therefore, they should show the calculation results of the electric-field distribution of the structures.
[3] The conditions of the simulation study are not described. What kind of simulation procedure is used to obtain results in section 4(e.g., DDT, FDTD, or FEM)? Which simulation parameters are employed?
[4] In Figure 2, why is the transmission of RQD smaller than others, especially GQD+RQD? Also, control experiments of the transmission spectra of samples without Ag-NPs, and solution of QDs are needed.
Round 2
Reviewer 1 Report
All the issues have been well addressed, and the manuscript is properply organized and well written. Thus, I recommend the manuscript to be accepted for publication in the current form.
Reviewer 2 Report
The revision has addressed all of my concerns. The revised manuscript is acceptable for publication in Nanomaterials